# Use of Platelet-Rich Fibrin in the Treatment of Grade 2 Furcation Defects: Systematic Review and Meta-Analysis

**DOI:** 10.3390/jcm9072104

**Published:** 2020-07-03

**Authors:** Francesco Tarallo, Leonardo Mancini, Luciano Pitzurra, Sergio Bizzarro, Michele Tepedino, Enrico Marchetti

**Affiliations:** 1Department of Life, Health and Environmental Sciences, University of L’Aquila, Piazzale Salvatore Tommasi 1, 67100 L’Aquila, Coppito, Italy; mancinileonardo94@gmail.com (L.M.); enrico.marchetti@univaq.it (E.M.); 2Department of Periodontology, Academic Centre for Dentistry Amsterdam (ACTA), University of Amsterdam and Vrije Universiteit Amsterdam, 1081 LA Amsterdam, The Netherlands; l.pitzurra@acta.nl (L.P.); s.bizzarro@acta.nl (S.B.); 3Department of Biotechnological and Applied Clinical Sciences, University of L’Aquila, Piazzale Salvatore Tommasi 1, 67100 L’Aquila, Coppito, Italy; m.tepedino@hotmail.it

**Keywords:** platelet-rich fibrin, regeneration, furcation, PRF, periodontal defect

## Abstract

In periodontitis patients, furcation defects are crucial sites to regenerate due to their complex anatomy. Various modern surgical techniques and use of biomaterials have been suggested in the literature. Among all, platelet-rich fibrin (PRF) has potential in tissue regeneration thanks to its role in the release of growth factors. Therefore, the purpose of this study was to evaluate the beneficial effect of the addition of PRF to open flap debridement (OFD) or as an adjuvant to other biomaterials such as bone grafts in the treatment of grade 2 mandibular furcation defects. Systematic research was carried out on the databases Medline, Scopus, Embase, and Cochrane Library and registered on PROSPERO (CRD42020167662). According to the PICO guidelines by Cochrane, randomized trials and prospective non-randomized trials were evaluated, with a minimum follow-up period of 6 months. The inclusion criteria were the absence of systemic diseases, non-smoking patients, and a population aged from 18 to 65 years. Vertical pocket probing depth (PPD), vertical clinical attachment level (VCAL), and gingival recession (REC) were the primary outcomes. Vertical furcation depth (VFD), and the percentage of bone defect fill (%v-BDF) were considered as secondary outcomes. A meta-analysis of the primary and secondary outcomes was performed. Publication bias was assessed through a funnel plot. Eighty-four articles were initially extracted. Eight randomized clinical trials were analyzed according to the exclusion and inclusion criteria. The Quality assessment instrument (QAI) revealed four articles at low risk of bias, one at moderate, and three at high risk of bias. The metanalysis showed significant data regarding PPD, VCAL, VFD and %v-BDF in the comparison between PRF + OFD vs. OFD alone. The adjunct of PRF to a bone graft showed a significant difference for VCAL and a not statistically significant result for the other involved parameters. In conclusion, the adjunctive use of PRF to OFD seems to enhance the periodontal regeneration in the treatment of grade 2 furcation defects. The combination of PRF and bone graft did not show better clinical results, except for VCAL, although the amount of literature with low risk of bias is scarce. Further well-designed studies to evaluate the combination of these two materials are therefore needed.

## 1. Introduction

The clinical management of furcation defects is still a crucial issue due to the position of the furcation and the irregular anatomy of the roots, which makes the biofilm virtually inaccessible for oral hygiene measures. The degree of furcation involvement represents per se a risk factor for tooth loss, with a heavy relative weight, next to several well-known patient-related factors such as age, gender, smoking habit, and diabetes [1,2]. The surgical treatment of these areas may involve a variety of reconstructive periodontal surgical techniques and materials [3]. The guided tissue regeneration (GTR) includes procedures attempting to regenerate the lost periodontal tissues when barrier materials are used to allow bone regrowth and new connective tissue attachment [3]. In this sense, the most commonly used materials are bone substitutes (autologous, allogenic, xenogenic, or synthetic by origin) alone or in combination with membranes (either resorbable or non-resorbable). Different types of membranes have been introduced ever since in order to provide space between the defect and the root surface, guide the proliferation of different strains of cells, and allow periodontal ligament cells, osteoblasts, and pericytes to repopulate the created space. However, the use of membranes requires specific considerations such as a more invasive flap, an additional surgery to remove the membrane (in case of non-resorbable membranes), and the possible exposure of the membrane that may compromise the results. In light of these considerations, the actual tendency is to prefer minimally-invasive-surgical procedures and to look for highly performant materials to limit the use of synthetic membranes, especially when treating contentive defects. Even though these regenerative materials are still used today, the introduction of biomimetic agents, such as enamel matrix derivatives, bone morphogenetic proteins, and platelets concentrates, raises new opportunities for better outcomes in periodontal treatment [4,5,6,7]. In very recent years, also the topical application of hyaluronic acid has further expanded the whole panorama of performant regenerative materials thanks to its favorable benefits in clinical attachment level (CAL) gain and vertical pocket probing depth (PPD) reduction when applied both in non-surgical and surgical periodontal therapy [8]. Promising results from the use of endogenous materials seem to overcome the limits of exogenous-crafted materials and enhance the regenerative potential of bone graft materials (Table 1).

Based on these considerations, interest in the use of platelet concentrates (PCs) for the treatment of many intraoral clinical conditions, including periodontal and furcation defects, is rising [15]. The rationale behind the preparation of platelet concentrates is that concentrated platelets and autologous growth factors could be collected from a whole blood sample through centrifugation and used in a surgical site to promote local healing [16,17,18]. In particular, natural fibrin is a biological three-dimensional matrix, enmeshed with platelets, cytokines, glycanic chains, and structural glycoproteins, which acts as a suitable network for breeding human periosteal cells, fibroblasts, and endothelial cell in tissue engineering [19]. They can be used alone or as a scaffold for other graft materials, favoring early tissue healing through the release of growth factors, chemokines, and cytokines. Two generations of PCs have succeeded in the last decades: first, platelet-rich plasma (PRP) and plasma rich in growth factors (PRGF); then, more recently, platelet-rich fibrin (PRF). The necessity of a second PCs generation emerged because several factors were shown to limit the use of PRP and PRGF. Their preparation requires the use of bovine thrombin or CaCl_2_ in addition to pro-coagulation factors, which has been shown as a potential inhibitor for early wound healing in in vitro studies. Furthermore, the solution must be centrifugated in two separate stages in order to increase platelet concentration, which makes it technically difficult and highly time-consuming. Third, this procedure does not allow the homogenous incorporation of leukocytes, which drops drastically the potential for cell mediator release and antibacterial function. Finally, the clinical potential for bone regeneration with PRP or PRGF is limited since they show a very short-term release of growth factors and a weak fibrin network [20,21,22]. PRF differs from its predecessors by several parameters, biological and technical. It has been shown that the relative amount, the variety, and the long-term release of cytokines is superior for the PRF in comparison to PRP and PRGF [21]. It is achieved with a simplified preparation, with no biochemical manipulation of blood and no request of anticoagulants or bovine thrombin. These features make this product easy to produce in a chair-side setup, with a standardized preparation procedure that is reproducible and has a low risk of errors [23].

The PRF preparation procedure requires a blood sampling from the patient that is subsequently centrifuged in a specific way to obtain the stratification of three different layers: red blood cell sediment at the bottom of the vial, platelet-poor plasma (PPP) located on the surface, and an intermediate layer, called the “buffy coat”, produced by the concentration of leucocytes and platelets [23]. The material obtained is cut with sterile scissors and it may be also manipulated to obtain membranes. A comparison of the three blood products was presented in the study by Giannini et al. [24] to show their main characteristics (Figure 1).

Based on the large quantity of platelets and cytokines that interacts with fibrin to form a hemostatic plug and to gradually release growth factors that stimulate wound healing, these types of biomaterials have been widely used in medical procedures, such as facial plastic surgery [25], skin ulcers [26], sinus-lift procedures [27], multiple gingival recession cases [28,29], and periodontal surgery, as well as in intrabony defect [30] and grade 2 furcation regeneration [31]. In the literature, various types of PRF preparations have been proposed based on different relative centrifugation forces (RCF) [32]. The leukocytes-PRF or L-PRF by Dohan et al. [23] is the most common one. However, it was demonstrated that the relative centrifugation force-max (RCFmax) used in the L-PRF is correlated with a higher weight of the pushed-away corpuscles (approximately 700× *g* 12 min). Therefore, the use of a low spin protocol, with a RCFmax around 200× *g* 8 min, leads to the greater retention of leukocytes, platelets, and other viable cells and to a more homogenous distribution enclosed with the fibrin clot [33]. This is the advanced-PRF or A-PRF procedure. This preparation showed a continuous and persistent release of growth factors from 1 week up to 28 days and possible enhancing effects on the healing process.

PRF is the most recent platelet concentrate and seems to be also the most performant of all PCs due to its mechanical and biological properties. It is conceivable to hypothesize that PRF could be a suitable material in the treatment of molar teeth with grade 2 furcation defects, which represent a great challenge to be maintained because of the poor clinical results after different surgical treatments [1]. Some systematic reviews that investigated the use of PCs are available in the scientific literature [15,34,35,36]. However, some studies pooled multiple types of PCs or considered all the possible applications of the PRF for regenerative dentistry, therefore possibly limiting the translational relevance of those results. Moreover, it is still unclear if the adjunct of PRF to a bone graft can further improve the clinical outcomes for the regenerative treatment of this anatomical area. In light of these considerations, the rationale behind a new review of the literature, focused only on the use of PRF and in grade 2 furcation treatment, seems to arise.

Therefore, the purpose of the present systematic review with meta-analysis was to evaluate the level of evidence that suggests the addition of PRF to either open flap debridement (OFD) or bone grafting (BG) procedures for the treatment of grade two furcation defects.

## 2. Materials and Methods

### 2.1. Protocol

This systematic review was conducted in accordance with the guidelines of the “Cochrane Handbook for Systematic Reviews of Interventions” and is reported following the PRISMA guidelines [37,38]. The methods of the analysis and inclusion criteria were specified in advance and registered on the international prospective register of systematic reviews, PROSPERO, with registration number CRD42020167662.

### 2.2. Eligibility Criteria

According to the formulated PICOS question, a framework developed to facilitate the literature search, this systematic review focused on all types of human studies (studies), on young and adult periodontal patients with grade 2 furcation defects in first and second mandibular molars with a probing depth of minimum 5 mm and a horizontal depth of 3 mm (population) that received the additional application of PRF to an OFD or a BG treatment (intervention), evaluating the amount of soft and hard tissue regeneration (outcome), compared to subjects treated with OFD or BG alone (comparator).

Randomized trials and non-randomized prospective studies, double blinded trials, and studies performed on either adults or adolescents regarding the use of PRF in furcation with a grade 2 defect treatment and a minimum follow-up period of 6 months were included. Based on the exclusion criteria, all studies conducted in vitro, animal studies, meta-analyses, case reports, mini reviews, conference proceedings, narrative revisions, and previous systematic reviews regarding the use of PRF in furcation defects treatment were not considered.

### 2.3. Information Sources and Search

The following scientific sources were searched without language and initial date restrictions, from the 3 June 2019 up to the 24 March 2020: MEDLINE via PubMed, Scopus, Web of Science and Cochrane Library. A search strategy was finalized utilizing MESH terms, Boolean operators, and free text terms and was the following: (platelet rich fibrin OR PRF) AND furcation. In addition, a manual search of the reference list of the potential studies was performed to retrieve additional articles. Duplicate articles were removed.

### 2.4. Study Selection

Eligibility was assessed by two authors (F.T. and L.M.), screening initially the title and abstract of the articles. Whenever the abstract was not clear if it should be included or not, full texts were then accessed. A PRISMA flowchart diagram for the study selection process is shown in Figure 2.

### 2.5. Data Extraction

Two authors (F.T. and L.M.) independently screened data (authors, year of publication, study design, sample size, sample composition by sex and age, presence of control group, method of assessment, follow-up period, inclusion and exclusion criteria, primary and secondary outcomes measures) from the selected studies. Any disagreement between the two authors was resolved by discussion and consensus. If no agreement could be reached, a third author (E.M.) was requested to judge.

### 2.6. Risk of Bias in Individual Studies

Quality and risk of bias assessment of the articles included in the review was performed using a quality assessment instrument (QAI) developed from a relevant article in the literature [39]. To evaluate the quality of the included studies, QAI was based on 7 stringent criteria. For the objective assessment of quality, a scoring system was incorporated. Each study earned one point if the answer to the corresponding criteria was positive, and no points if the answer was negative or unclear. Low risk of bias was assigned to a study when random allocation, defined inclusion/exclusion criteria, blinding to patient and examiner, balanced experimental groups, identical treatment between groups (except for the intervention), and reporting of follow-up were present. Moderate risk of bias was attributed to the studies that met six of these seven criteria. If two or more of these seven criteria were absent, the study was considered to have a high risk of bias. Any disagreement between the two authors was solved by the intervention of a third experienced reviewer (E.M.).

### 2.7. Summary Measures

A narrative synthesis was performed by illustrating the results from individual studies according to the protocol. Vertical probing depth of the pocket (PPD), vertical clinical attachment level (VCAL), and gingival recession (REC) were considered as primary outcomes. Vertical furcation depth (VFD) and percentage of vertical bone fill (%v-BDF) were treated as secondary outcomes.

When three or more studies showed a homogeneous methodology, a meta-analysis was performed to obtain an estimate of the effect size. The meta-analysis was performed with Review Manager (RevMan version 5.3. Copenhagen, Denmark: The Nordic Centre, The Cochrane Collaboration, 2014) [40], a free, cross-platform, open-source program for meta-analysis of efficacy and diagnostic test accuracy studies.

Split-mouth and parallel-arm studies evaluating OFD procedures (PRF + OFD vs. OFD alone) and BG procedures (PRF + BG vs. BG alone) were analyzed separately. All results were combined using a random effects model, and the mean difference and 95% confidence interval (CI) were used as effect measures for all primary and secondary outcomes. A *p* value < 0.05 was considered as statistically significant.

### 2.8. Risk of Bias Across Studies

Heterogeneity among the studies was assessed by using the Cochrane Q test, and the I2 test was performed to measure the proportion of inconsistency in the combined estimates due to between-study heterogeneity. Low heterogeneity was attributed with I2 values lower than 30%, moderate heterogeneity with values of 30% to 60%, and substantial heterogeneity with values of over 60%. Publication bias (including small-study effects) was assessed through visual inspection of asymmetry on a funnel plot [41].

## 3. Results

### 3.1. Study Selection

An electronic database search provided a total of 83 results. Twenty-six articles were collected from PubMed, 31 from Scopus, and 26 from Web of Science. One additional article was retrieved through manual search. After checking for duplicates, 46 entries remained to be screened. Of these, 24 titles were discarded because they were clearly not relevant, and 22 abstracts were screened. Seven abstracts were discarded due to the methodology not corresponding to the inclusion criteria; thus 15 full text papers were suitable for detailed examination; six articles were discarded according to the exclusion criteria after full text screening, and one article was excluded because some inconsistencies between the data reported in the tables and those reported in the text were found. In conclusion, eight studies were included in the systematic review [31,42,43,44,45,46,47,48].

### 3.2. Study Characteristics

Some heterogeneity in study design (split-mouth design and parallel design) was observed. All studies were randomized clinical trials (RCTs) and two of them presented a split-mouth design to avoid the effect of a natural variation between different individuals. One study did not mention the gender distribution within the sample [48]. All studies were homogeneous in excluding endodontically-treated teeth, mobility ≥ grade 2, and smokers and patients with systemic diseases that could affect the regenerative potential, like diabetes. All the studies considered the pre-surgical therapy phase, consisting of supra- and sub-gingival debridement, scaling and root planing, and plaque control instructions. The studies were categorized based on the population characteristics as described in Table 2, and on the clinical and laboratory parameters measured (Table 3) and their findings. The PRF-centrifugation protocol was not homogeneous across the included studies and neither was the RCFmax (400× *g* for 10 [42] or 12 [45,47] minutes and 3000 rpm for 10 [31,43,46,48] or 12 [44] minutes). Non-standard types of centrifuges were used, and therefore the RCF was not comparable among them. The follow-up for re-evaluating the clinical and radiographic parameters was 6 or 9 months. Clinical measurements were made with customized acrylic stents and a UNC-15 periodontal probe for soft tissue parameters or with a Nabers probe for furcation measures in each study. Periapical radiographs of the operated site were often taken for hard tissue assessment using customized acrylic bite-blocks and the paralleling technique. Only two studies [45,48] performed pre- and post-evaluation TC Cone Beam to overcome the limits of intraoral periapical radiographs. One study [44] recorded data with a surgical re-entry. Four studies [31,42,43,45] performed a proper pre-surgical calibration of the examiner. No study performed a minimally invasive surgery, but buccal and lingual sulcular incisions were made, and the muco-periosteal flaps were reflected. However, all the studies affirmed that the treatment protocol emphasized the principles of careful soft tissue handling, wound stability, and infection control for the post-surgery phase.

### 3.3. Risk of Bias in Individual Studies

Assessment of the risk of bias using the previous QAI tool, revealed that four articles [31,42,43,48] were at a low risk of bias, one article [47] was at moderate risk of bias, and three articles [44,45,46] were at a high risk of bias (Table 4).

### 3.4. Risk of Bias Across Studies

Heterogeneity was moderate or large for most of the comparisons, except for the estimate of the mean difference between PRF + BG and BG alone, which showed a low I2 value. Visual inspection of the funnel plot revealed a certain degree of asymmetry. However, studies with null or negative effects were also included; therefore, it is likely that the heterogeneity observed was due to other factors than selective reporting of positive outcomes (Figure 3).

### 3.5. Qualitative Synthesis

PRF has been used in many studies investigating periodontal regeneration of grade 2 furcation defects. Five articles [31,42,43,45,47] compared the effect of PRF to OFD alone. In these studies, the use of PRF led to a significant improvement in VCAL and reduction in PPD when compared to controls.

Three authors [44,46,48] evaluated the effectiveness of the addition of PRF to other bone grafting materials compared to BG procedures without PRF. In particular, Lohi et al. considered the efficacy of the addition of PRF to bioactive ceramic composite granules (BCCG) compared to BCCG alone. A significant improvement was observed for all the measured parameters in the PRF + BCCG group compared to the BCCG group over six months. Furthermore, an intergroup comparison revealed a statistically significant increase in radiographic bone density at the furcation defect in the test group, indicating that PRF may enhance the regenerative capacity of bioactive ceramic composite granules [46]. On the other hand, Rani et al. evaluated the efficacy of the combination of PRF with beta-three calcium phosphate (β-TCP) versus β-TCP alone, showing a statistically significant difference for overall parameters from their baseline, but the intergroup comparison revealed a non-significant difference for both hard and soft tissue parameters. Basireddy et al. considered the comparison of PRF with demineralized freeze-dried bone allograft (DFDBA) for the treatment of grade 2 furcation defects. The clinical parameters at 6 months showed a greater improvement but no significant difference in the intergroup comparison. Furthermore, this is the only study where the use of periodontal dressing was not mentioned in covering the surgical site.

In all the studies, antibiotics (mainly amoxicillin 500 mg) and pain killers were prescribed for the post-surgical week in association with mouth rinses of chlorhexidine digluconate at 0.12% [31,42,43] or 0.20% [44,45,46,47]. Moreover, an overall improvement both in plaque index and gingival index was recorded in all the studies at follow-up in comparison to the baseline, which was statistically significant. No significant difference was observed between the study groups and the controls for both the parameters. No adverse effect or reactions against the material were observed, although patient-centered outcomes were not considered in any of the included studies.

### 3.6. Meta-Analysis

All the eight selected studies were included in the meta-analysis of the primary outcomes: five trials were comparing PRF + OFD to OFD alone, and three studies were comparing PRF + BG to BG alone. Forest plots of the secondary outcomes (VFD, %v-BDF) analysis were only performed for the PRF + OFD vs. OFD group, since no sufficient data were available for the PRF + BG vs. BG group.

(1)PRF + OFD vs. OFD alone:

The comparison between PRF + OFD and OFD alone showed a statistically significant difference for two out of three primary outcomes. PPD and VCAL estimates suggested a positive effect of the addition of PRF to OFD procedures (1.73 mm and 1.42 mm respectively, *p* < 0.05). On the other hand, the use of PRF seems to have no significant effect on REC levels (0.14 mm, *p* = 0.40) (Figure 4). Regarding the secondary outcomes, the adjunctive use of PRF to OFD showed a favorable and statistically significant effect on both VFD and %v-BDF (1.54 mm and 37.61% respectively, *p* < 0.05) (Figure 5).

(2)PRF + BG vs. BG alone:

No differences between PRF + BG and BG alone were found regarding PPD and REC (0.30 mm and 0.30 respectively, *p* > 0.05), while the VCAL estimates were suggesting a positive effect of the addition of PRF to BG procedures (0.71 mm, *p* < 0.05) (Figure 6).

## 4. Discussion

### 4.1. Summary of Evidence

Advantages in using PRF + OFD versus OFD alone:

Several studies [31,42,43,45,47] confirmed the beneficial effect of PRF in periodontal regeneration compared to OFD only. Because among the included studies three were at low risk of bias, one was at moderate risk, and one at high risk of bias, precautions may be required while evaluating the present results. Nevertheless, the advantages in using PRF due to its reported good biological effects, low costs, and ease of preparation may also be considered. Hence, the application of PRF in the treatment of grade 2 furcation defects may be largely encouraged and preferred with respect to the approach of a solely OFD.

Advantages in using PRF + BG versus BG alone:

It has been further verified whether the application of additional material to the PRF is associated with better results in VCAL. As described by several studies [44,46,48], an overall improvement in most of the clinical parameters is reported, and the increase is due to a synergistic effect of the BG material and the additional PRF. However, the results may differ depending on the type of material used.

Lohi et al. [46] applied BCCG, a graft material composed of 50% bioactive glass and 50% hydroxyapatite. Its bioactivity allows a tight fixation of the granules resulting from direct bonding to the living bone. Moreover, the presence of stem cells promotes the differentiation into periodontal cellular elements, thus boosting the regenerative potential of PRF. The ability of the PRF to enhance the regenerative capacity of BCCG is also highlighted by a great increase in radiographic bone density. However, the examiners involved in the trial were not blinded, which may have introduced significant bias in the assessment of the parameters.

Rani et al. [44] used β-TCP. This material is fully resorbed and replaced by natural bone within a reasonable period. It is considered an attractive bone substitute due to its biocompatibility, biological safety, virtually unlimited availability, ease of sterilization, and long shelf life. β-TCP represents a good balance among absorption, degradation, and new bone formation through the preservation of the structural stability and the release of a large quantity of calcium (Ca^2+^) and sulphate (SO4^2−^) ions, fundamental inorganic molecules involved in bone turnover. β-TCP, with the addition of PRF, has been shown to be an effective regenerative material in the management of grade 2 furcation, displaying a great reduction in vertical and horizontal pocket depth and gain in CAL. However, the results of this study should be considered very cautiously, as Rani et al. obtained a higher PPD reduction in the group treated with β-TCP alone. The authors explained this result as due to an initially difficult adaptation of the PRF membrane, which hindered the proper placement and stabilization of the graft. Furthermore, both the previous studies showed high risks of bias.

Basireddy et al. [48] combined PRF with DFDBA to achieve a synergistic effect. DFDBA provides an osteoconductive surface and, in addition, it acts as a source of osteoinductive factors. Therefore, it stimulates mesenchymal cell migration, attachment, and osteogenesis when implanted in well-vascularized bone. Otherwise, it induces endochondral bone formation when implanted in tissues that would not form bone. DFDBA contains bone morphogenic proteins (BMPs) such as BMP 2, 4, and 7, which help stimulate osteoinduction. The addition of PRF to DFDBA seemed to slightly favor soft-tissue healing but did not affect the bone volume. The lack of difference in radiographic parameters is attributed to the use of DFDBA in both the groups, which may have overcome the effect of PRF. The methodology of this study was at low risk of bias.

In conclusion, the results of this systematic review are in accord with several other systematic reviews [15,35,36] that previously analyzed the effects of PCs for the surgical treatment of furcation defects, reporting positive effects when PRF is combined with OFD. In particular, the systematic review conducted by Panda et al. reported outcomes regarding the comparison between PRF + OFD and OFD, similar to those of the present meta-analysis, which conversely included an additional study. On the other hand, differences were observed for the comparison between PCs + BG and BG alone because they mixed into the meta-analysis the results of two types of platelet concentrates (PRF and PRP) due to the similarities in their outcomes. The assimilation of both types of PCs may have introduced a bias, because a vast literature has previously shown the differences in handling, ease of preparation, and potential healing effects of the different autologous platelet concentrates [19,21,49,50,51], while the present systematic review considered exclusively the PRF among all the platelet derivates. However, the results of the present meta-analysis showed that the addition of PRF to a bone graft does not provide an advantage compared to the use of bone grafting alone for the surgical treatment of grade 2 furcation defects, at least for the graft materials used in the included studies, although some slight benefits were solely achieved for VCAL.

### 4.2. Limitations

The included studies showed some heterogeneity in centrifugation preparation, in the handling of PRF, and in the post-surgery measurements. The relatively small sample size and the short-term follow-up may be not long enough to draw definitive conclusion on tooth survival. The age range of the included samples was quite broad (25–60 years), and this could have affected the present results, since some studies reported better outcomes for the use of PRF in young patients compared to elder ones [52]. A histomorphometric analysis would have been helpful to confirm the newly formed tissues but was not included in the selected studies, and indeed it is seldom feasible. Maxillary furcation defects were not included, limiting the results to mandibular molars. Further randomized clinical studies with a long-term follow-up are therefore needed to determine the prognosis and survival rate of the treated teeth.

## 5. Conclusions

Basing on the results of the present systematic review, the following implications can be summarized:-All the studies found favorable outcomes in adding PRF to an open flap debridement in terms of PPD and VCAL;-Positive effects on both hard- (VFD, %v-BDF) and soft-tissue (PPD, VCAL, REC) healing were associated with the use of PRF for the surgical treatment of grade 2 furcation defects;-Future studies will need to evaluate the real regenerative potential through histological assessment. Hence, based on the current evidence, the process may be solely defined as tissue repair;-The comparison among the tested bone grafts did not show a clear superiority of one method over the others nor a statistical significance in adding PRF to the bone graft except for VCAL;-To achieve predictable regenerative outcomes in the treatment of furcation defects, adverse systemic and local factors (diabetes, smoking, oral hygiene, vitality, and mobility of the tooth) should be evaluated and controlled, if possible;-Patient-centered outcomes might be considered in evaluating the risk/benefit ratio;-Long-term follow-up is required to evaluate the tooth survival rate and to guide therapeutic prognosis of teeth presenting furcation defects.

Within its limits, PRF demonstrated better results than OFD alone in furcation treatment while its adjunction to BG seems to have less advantages. However, more well-designed RCTs with low risk of bias are needed to clarify its potential role in combination with BGs. In order to obtain a major homogeneity among the data, future studies may clearly express the relative centrifugation force (RCF) in their protocol and consider the same clinical (PPD, VCAL, REC) and radiographical (VFD, %v-BDF) parameters to being measured.

## Figures and Tables

**Figure 1 jcm-09-02104-f001:**
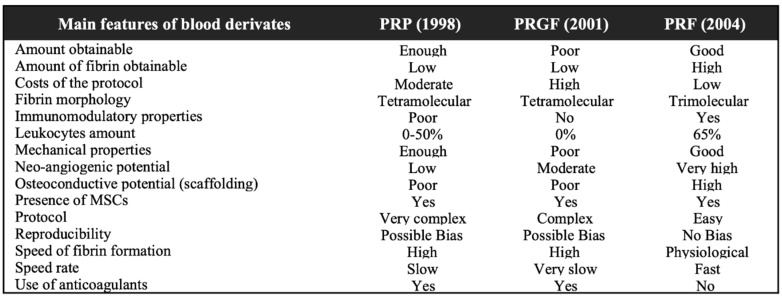
Overview table comparing the main three blood products and their characteristics. Table modified from Giannini et al. [24] PRP, platelet-rich plasma; PRGF, plasma rich in growth factors; PRF, platelet rich fibrin.

**Figure 2 jcm-09-02104-f002:**
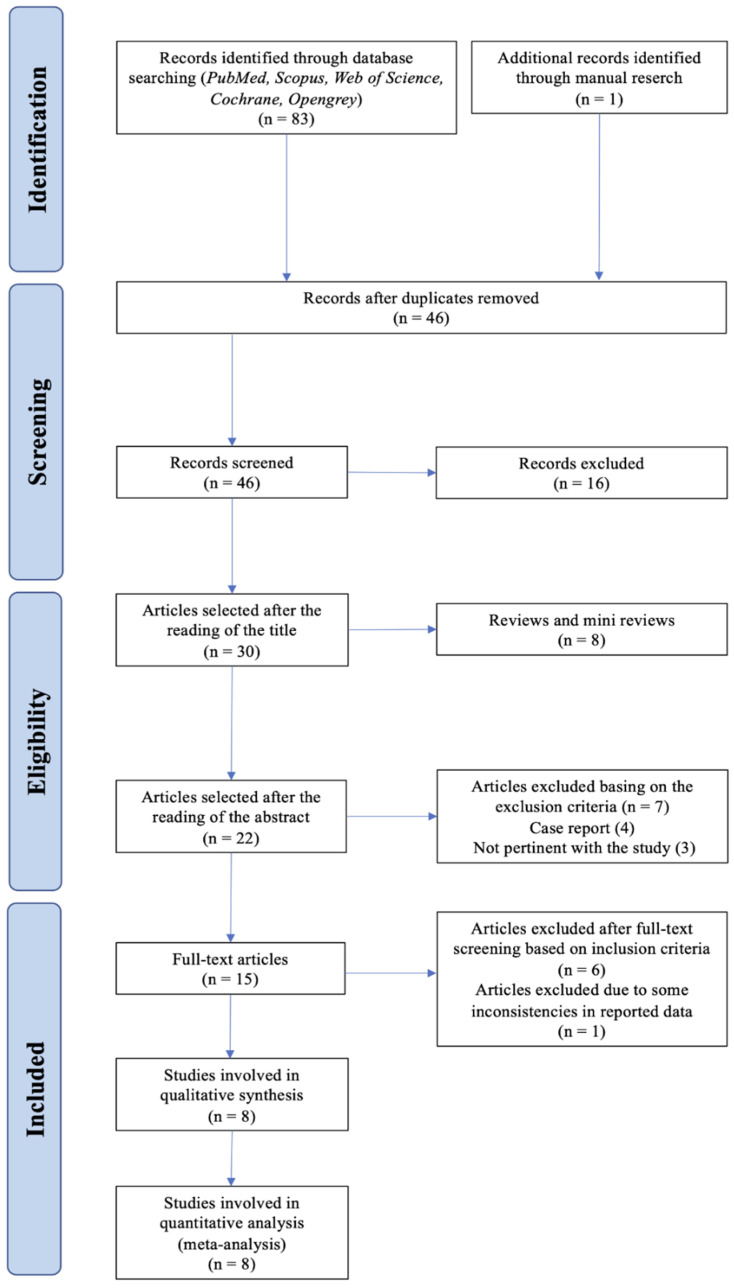
PRISMA flowchart for study selection.

**Figure 3 jcm-09-02104-f003:**
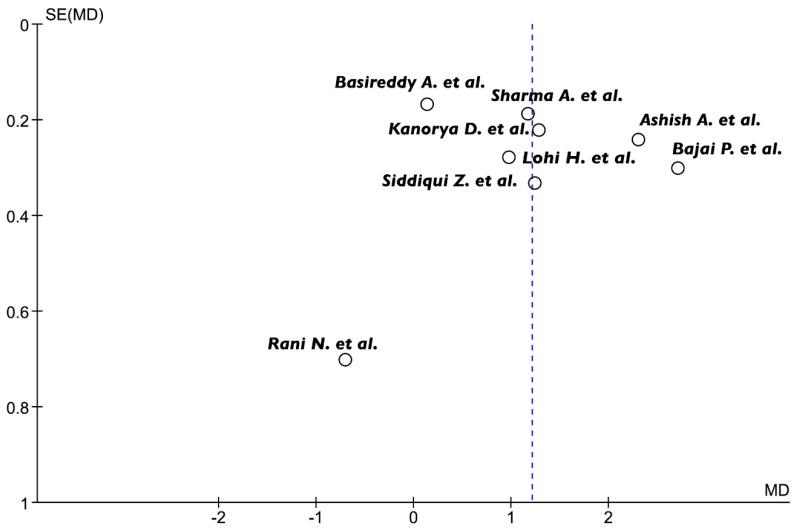
Funnel-plot analysis revealed some heterogeneity among the studies. SE, standard error; MD, mean difference.

**Figure 4 jcm-09-02104-f004:**
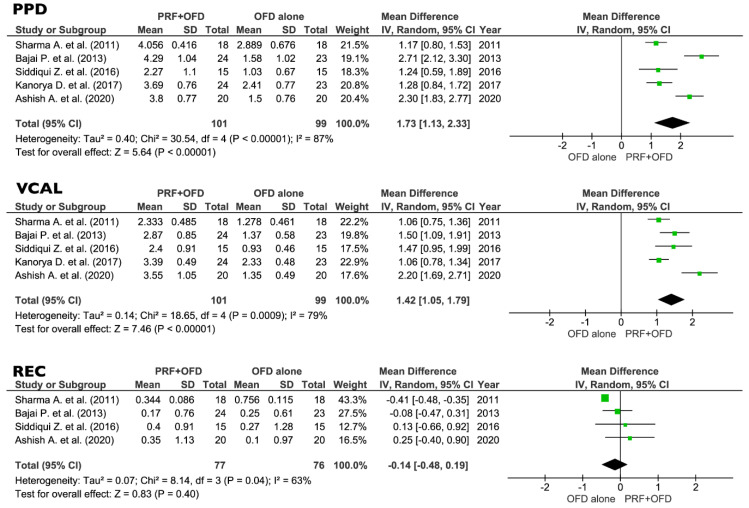
Effect of PPD reduction, VCAL gain, and REC, respectively, in the group PRF + OFD vs. OFD alone. PPD, vertical pocket probing depth; VCAL, vertical clinical attachment level; REC, gingival recession; PRF, platelets rich fibrin; OFD, open flap debridement; SD, standard deviation; CI, confidence interval.

**Figure 5 jcm-09-02104-f005:**
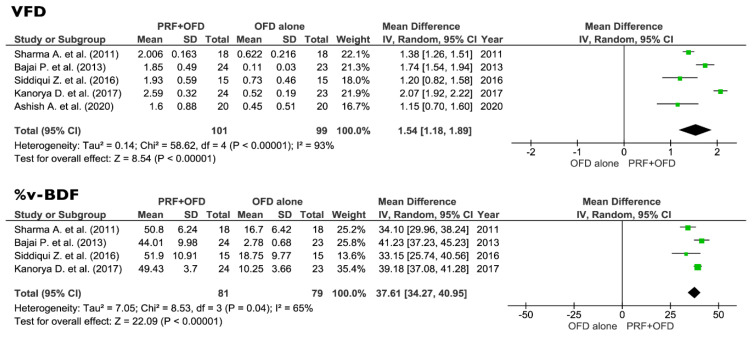
Effect of HFD gain, VFD gain and %v-BDF respectively in the group PRF + OFD vs. OFD alone. VFD, vertical furcation depth; v-BDF, percentage of bone defect fill; PRF, platelets rich fibrin; OFD, open flap debridement; SD, standard deviation; CI, confidence interval.

**Figure 6 jcm-09-02104-f006:**
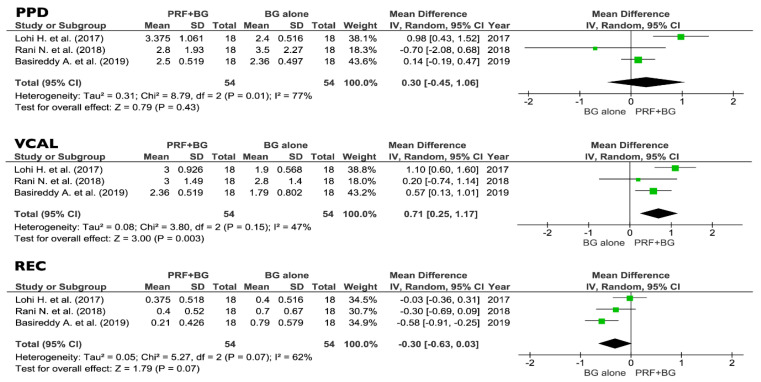
Effect of PPD reduction, VCAL gain, and REC, respectively, in the group PRF + BG vs. BG alone. PPD, vertical probing depth; VCAL, vertical clinical attachment level; REC, gingival recession; PRF, platelets rich fibrin; BG, bone graft; SD, standard deviation; CI, confidence interval.

**Table 1 jcm-09-02104-t001:** Negative features of bone-graft materials.

***Autograft***
The graft amount may be insufficient
Association with 8–38% risk of complications, i.e., infection, hematoma, nerve injury, cosmetic disadvantages, pain, and morbidity of the donor site [9]
Irregular rate of resorption of the autologous bone may require a secondary corrective surgery [10]
Additional surgical sites
More time requested at the chair
***Allograft***
Variable host immune response [11]
Limited supplies [12]
Culturally unacceptable in some countries
High cost
***Xenograft***
Delays the early bone formation [13]
Lack of sufficient intrinsic osteconductivity [14]
Culturally unacceptable in some countries
High cost
***Alloplastic Materials***
Lack of osteoinductive properties
Delayed healing
High cost

**Table 2 jcm-09-02104-t002:** Characteristics of the studies included in the systematic review.

Authors (year)	Study Design	No. of Patients	No. of Surgical Sites	Study Group(s)	Control Group(s)	Mean Age (years)	n (M)	n (F)
Ashish A. et al. (2020)	Randomized clinical trial of parallel design	46	60	PRF + OFD (20 sites); PRF + DFDBA + OFD (20 sites)	OFD (20 sites)	30–65 (mean age 48 ± 15 years)	20	26
Bajaj P. et al. (2013)	Randomized, double blinded, controlled clinical trial of parallel design	37	72	PRF = 12 (24 sites); PRP = 13 (25 sites)	OFD = 12 (23 sites)	39.4	22	20
Basireddy A. et al. (2019)	Randomized, double blinded, controlled clinical study, split mouth study	14	28	DFDBA + PRF (14 sites)	DFDBA alone (14 sites)	30-50	-	-
Kanoriya D. et al. (2017)	Randomized, double blinded, controlled clinical trial of parallel design	72	72	PRF = 24 (24 sites); PRF + 1% ALN gel = 25 (25 sites)	OFD = 23 (23 sites)	30–50 (mean age 38 years)	36	36
Lohi H. et al. (2017)	Randomized clinical trial of parallel design	16	18	PRF + BCCG (8 sites)	BCCG alone (10 sites)	25–65 (mean age 43.05 years)	12	4
Rani N. et al. (2018)	Randomized clinical trial of parallel design	20	20	β-TCP alone (10 sites)	β-TCP + PRF = 10 (10 sites)	25–50	-	-
Sharma A. et al. (2011)	Randomized, double blinded, controlled clinical study, split mouth study	18	36	PRF + OFD = 9 (18 sites)	OFD = 9 (18 sites)	34.2	10	8
Siddiqui Z.R. et al. (2016)	Randomized clinical trial of parallel design	31	45	PRF + OFD (15 sites); β-TCP + OFD (15 sites)	OFD alone (15 sites)	-	24	7

PRF, platelet rich fibrin; PRP, platelet rich plasma; OFD, open flap debridement; ALN, alendronate; BCCG, bioactive ceramic composite granules; DFDBA, demineralized freeze-dried bone allograft; β-TCP, beta tricalcium phosphate; MF, metformin.

**Table 3 jcm-09-02104-t003:** Clinical data of the studies included in the systematic review.

Authors (year)	Clinical Parameters	Follow-up	Pre-Surgical Procedure	Surgical Approach	Post-Surgical Management	Centrifugation Speed	*p* Value
Ashish A. et al. (2020)	PI, GI, PPD, HPD, CAL, REC, VFD, HFD, FW	9 months	Phase-I periodontal therapy + plaque control instructions	PRF: clot + membrane; OFD: curettes and ultrasonic instruments	Amoxicillin (500 mg) + Ibuprofen (800 mg) + 0.20% chlorhexidine gluconate	400× *g* for 12 min	*p* < 0.05
Bajaj P. et al. (2013)	PI, SBI, PPD, VCAL, RHCAL, REC, VFD	9 months	Phase-I periodontal therapy + plaque control instructions	PRF: clot + membrane; OFD: curettes and ultrasonic instruments	Amoxicillin (500 mg) + Ibuprofen (800 mg) + 0.12% chlorhexidine gluconate	400× *g* for 10 min	*p* < 0.001
Basireddy A. et al. (2019)	PI, SBI, PPD, VCAL, HCAL, REC, VFD, HFD	6 months	Phase-I periodontal therapy + plaque control instructions	PRF: clot mixed with DFDBA; BG: DFDBA alone	Amoxicillin (500 mg) + Combination of diclofenac and paracetamol + Chlorhexidine gluconate	3000 rpm for 10 min	*p* < 0.05
Kanoriya D. et al. (2017)	PI, SBI, PPD, VCAL, HCAL, VFD	9 months	Phase-I periodontal therapy + plaque control instructions	PRF: clot + membrane; OFD: curettes and ultrasonic instruments	Amoxicillin (500 mg) + Metronidazole (500 mg) + Ibuprofen (800 mg) + 0.12% chlorhexidine gluconate	3000 rpm (approximately 400× *g*) for 10 min	*p* < 0.05
Lohi H. et al. (2017)	PI, GI, PPD, VCAL, HCAL, REC, VFD, HFD	6 months	Phase-I periodontal therapy + plaque control instructions	PRF: clot mixed with BCCG + membrane; BG: BCCG alone	Amoxicillin (500 mg) + Ibuprofen (400 mg) + Paracetamol (325 mg) + 0.20% chlorhexidine gluconate	3000 rpm (approximately 400× *g*) for 10 min	*p* < 0.05
Rani N. et al. (2018)	PPD, VCAL, REC, VFD, HFD	6 months	Phase-I periodontal therapy + plaque control instructions	PRF: β-TCP + membrane; BG: β-TCP alone	Novamox LB (250 mg) + Ibuprofen (400 mg) + 0.20% chlorhexidine gluconate	3000 rpm for 12 min	*p* < 0.05
Sharma A. et al. (2011)	PI, SBI, PPD, VCAL, HCAL, REC, VFD	9 months	Phase-I periodontal therapy + plaque control instructions	PRF: clot + membrane; OFD: curettes and ultrasonic instruments	Amoxicillin (500 mg) + Ibuprofen (800 mg) + 0.12% chlorhexidine gluconate	3000 rpm (approximately 400× *g*) for 10 min	*p* < 0.05
Siddiqui Z.R. et al. (2016)	PI, GI, PPD, VCAL, HCAL, REC, VFD, HFD, FW	6 months	Phase-I periodontal therapy + plaque control instructions	PRF: only clot; OFD: curettes and ultrasonic instruments	Amoxicillin (500 mg) + Ibuprofen (400 mg) + 0.20% chlorhexidine gluconate	2700 rpm (approximately 400× *g*) for 12 min	*p* < 0.05

PI, plaque index; GI, gingival index; SBI, sulcus bleeding index; PPD, vertical probing depth of the pocket; VCAL, vertical clinical attachment level; HCAL, horizontal clinical attachment level; REC, gingival recession; GMP, gingival margin position; VFD, vertical furcation depth; HFD, horizontal furcation depth; FW, furcation width.

**Table 4 jcm-09-02104-t004:** Risk of bias summary: judgement about each risk of bias item presented across all included RCTs.

	Ashish Agarwal et al. (2020)	Bajaj P. et al. (2013)	Basireddy A. et al. (2019)	Kanoriya D. et al. (2017)	Lohi H. et al. (2017)	Rani N. et al. (2018)	Sharma A. et al. (2011)	Siddiqui Z.R. et al. (2016)
**Random allocation**	Y	Y	Y	Y	Y	Y	Y	Y
**Inclusion/exclusion criteria clearly defined**	Y	Y	Y	Y	Y	Y	Y	Y
**Blinding of participants**	Unclear	Y	Y	Y	Unclear	Unclear	Y	Unclear
**Blinding of examiners**	Y	Y	Y	Y	N	Unclear	Y	Unclear
**Balanced experimental groups**	Y	Y	Y	Y	Y	Y	Y	Y
**Identical treatment between the groups**	Y	Y	Y	Y	Y	Y	Y	Y
**Reporting of follow-up**	Y	Y	Y	Y	Y	Y	Y	Y
**Total**	**6 on 7**	**7 on 7**	**7 on 7**	**7 on 7**	**5 on 7**	**5 on 7**	**7 on 7**	**5 on 7**

RTC, randomized clinical trials; one point was attributed when the answer was Yes. If the answer was No or Unclear, no more points were attributed. Assessment of the risk of bias: low risk, moderate risk, high risk of bias.

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
