# Peer review of "Use of Platelet-Rich Fibrin in the Treatment of Grade 2 Furcation Defects: Systematic Review and Meta-Analysis"

_jcm, 2020, doi:10.3390/jcm9072104_

Round 1

Reviewer 1 Report

This well written systematic review aims to review the potential clinical benefits of using platelet-rich fibrin (PRF) with or without bone grafts in regeneration procedures to treat grade 2 furcation mandibular defects.

The authors have overall respected PRISMA guidelines to report this systematic review and meta-analysis.

However, a few issues need to be addressed by the authors before this manuscript is ready for publication:

Overall comments:

  • The English grammar and syntax is good overall and only few punctuations and errors were detected.
  • The overall structure of the manuscript is clear, concise and to the point.
  • Item #2 (abstract) of PRISMA guidelines was not respected.
  • The bibliography section respects the journal policy and structure.
  • Only minor corrections are needed before publication.

Abstract:

  • The objectives, study eligibility criteria and participants main characteristics are not clearly stated in the abstract: the PICO question could be presented to cover these missing informations.
  • Line 15: remove “still” in “Furcation defects are still crucial sites…”
  • Line 26: were “favorable effects” significant?
  • Line 30: add “mandibular” in “…regeneration in the treatment of grade 2 furcation “mandibular” defects.”
  • Line 31: add a coma after “…better clinical results, …” and after “…except for VCAL,…”
  • Line 31: add “with low risk of bias” after “… although the amount of literature “with low risk of bias” is scarce.”

Introduction:

  • Line 53: add a coma after “…the use of synthetic membranes,…”
  • Line 70: add a coma after “…materials, ….”
  • Line 71: replace “years” by “decades”
  • Line 75: add a coma after “…factors,…
  • Line 75: replace “possibly” by “potential”
  • Line 85: replace “on” in “on a chair-side setup” with “in”
  • Line 97: replace “platelets-rich plasma” by “platelet-rich plasma”
  • Line 103: replace “furcation treatment” by “furcation regeneration”
  • Line 104: add a coma after “In the literature,…”
  • Line 104: remove coma after “… have been proposed…”
  • Line 110: add “procedure” after “This is the Advanced-PRF or A-PRF procedure.”
  • Line 121: replace “this anatomical area” by “the furcation defect” to enhance clarity

Materials and Methods:

  • Eligibility criteria: was there a minimal length of follow-up? Not clearly stated here
  • Eligibility criteria: please state that in was in mandibular molars only.
  • Figure 2 (PRISMA flowchart): please add last box titled: “studies involved in quantitative analysis (meta-analysis)
  • Line 169: who was the third author arbitrating in case of disagreement?

Discussion:

  • Line 346: add a sentence about the fact that examiners were not blinded to study group assignments in Lohi et al study, which created significant bias in parameters assessment

Bibliography:

  • Line 437,438: justify to the left
  • Line 557,558: justify to the left

Author Response

Response to Reviewer 1 Comments

Point 1: The objectives, study eligibility criteria and participants main characteristics are not clearly stated in the abstract: the PICO question could be presented to cover these missing informations.

Response 1: Thank you for your comment. We modified the abstract adding the information you requested to include the missing data.

Point 2: Line 15: remove “still” in “Furcation defects are still crucial sites…”

Line 26: were “favorable effects” significant?

Line 30: add “mandibular” in “…regeneration in the treatment of grade 2 furcation “mandibular” defects.”

Line 31: add a coma after “…better clinical results, …” and after “…except for VCAL,…”

Line 31: add “with low risk of bias” after “… although the amount of literature “with low risk of bias” is scarce.”

Response 2: Point taken, we amended the Abstract’s test as suggested.

Point 3: Line 53: add a coma after “…the use of synthetic membranes,…”

Line 70: add a coma after “…materials, ….”

Line 71: replace “years” by “decades”

Line 75: add a coma after “…factors,…

Line 75: replace “possibly” by “potential”

Line 85: replace “on” in “on a chair-side setup” with “in”

Line 97: replace “platelets-rich plasma” by “platelet-rich plasma”

Line 103: replace “furcation treatment” by “furcation regeneration”

Line 104: add a coma after “In the literature,…”

Line 104: remove coma after “… have been proposed…”

Line 110: add “procedure” after “This is the Advanced-PRF or A-PRF procedure.”

Line 121: replace “this anatomical area” by “the furcation defect” to enhance clarity

Response 3: Point taken, we amended the Abstract’s test as suggested.

Point 4: Eligibility criteria: was there a minimal length of follow-up? Not clearly stated here

Response 4: Thank you for your comment. The minimum length of follow-up was 6 months. We added a sentence to page 4 line 154 to include this information.

Point 5: Eligibility criteria: please state that in was in mandibular molars only.

Response 5: Thank you for your suggestion, we modified the eligibility criteria accordingly.

Point 6: Figure 2 (PRISMA flowchart): please add last box titled: “studies involved in quantitative analysis (meta-analysis)

Response 6: we added the last box in Figure 2 as requested, thank you for the suggestion.

Point 7: Line 169: who was the third author arbitrating in case of disagreement?

Response 7: The third author was E.M. We forgot to write the first letters, but we have added them right now.

Point 8: Line 346: add a sentence about the fact that examiners were not blinded to study group assignments in Lohi et al study, which created significant bias in parameters assessment

Response 8: Point taken, we added a sentence in the Discussion as requested, thank you for the suggestion.

Point 9: Line 437,438: justify to the left; Line 557,558: justify to the left

Response 9: Point taken, we formatted the text as suggested.

Reviewer 2 Report

In the manuscript entitled: “Use of platelet-rich fibrin in the treatment of grade 2 furcation defects: systematic review and meta- analysis” the authors performed a systematic review with meta-analysis in order to evaluate the level of evidence that suggests the addition of PRF to either open flap debridement (OFD) or bone grafting (BG) procedures for the treatment of grade 2 furcation defects.

The authors found, on 8 randomized clinical trials that were analyzed basing on the inclusion and exclusion criteri, that the QAI tool revealed four articles at low risk of bias, one at moderate and three at high risk of bias. The metanalysis showed favorable effects regarding PPD, VCAL, VFD and %v-BDF in the comparison between PRF (platelet-rich fibrin) + OFD (open flap debridement) Vs OFD alone. The adjunct of PRF to a bone graft showed a significant difference for VCAL and a not statistically significant result for the other parameters.

The authors concluded that the adjunctive use of PRF to OFD seems to enhance the periodontal regeneration in the treatment of grade 2 furcation defects. The combination of PRF and bone graft did not guarantee better clinical results except for VCAL

Major comments:

In general, the idea and innovation of this study, regards the systematic review of platelet-rich fibrin in the treatment of grade 2 furcation defects is interesting, because the role of these outcomes are validated but further studies on this topic could be an innovative issue in this field could be open an innovative matter of debate in literature by adding new information. Moreover, there are few reports in the literature that studied this interesting topic with this kind of study design.

The study was well conducted by the authors; However, there are some concerns to revise that are described below.

The introduction section resumes the existing knowledge regarding PRF on furcation defects.

However, as the importance of the topic, the reviewer strongly recommends to update the literature through read, discuss and cites in the references with great attention all of those recent interesting articles, that helps the authors to better introduce and discuss the aim of the study in light of the some factors related with others adhuvants, such as hyaluronic acid

The authors should be better specified, at the end of the introduction section, the rational of the study. In the material and methods section, should better clarify the outcome assessment and the period of screening of the articles.

The conclusion should reinforce in light of the discussions.

In conclusion, I am sure that the authors are fine clinicians who achieve very nice results with their adopted protocol. However, this study, in my view does not in its current form satisfy a very high scientific requirement for publication in this journal and requests a revision before publication.

Minor Comments:

Abstract:

  • Better formulate the introduction section by better describe the background

Introduction:

  • Page 3, Line 71: please add the relative sentence

Discussion

  • Please add a specific sentence that clarifies the results obtained in the first part of the discussion
  • Page 14 last paragraph of discussion: Please reorganize this paragraph that is not clear

Author Response

Response to Reviewer 2 Comments

Point 1: However, as the importance of the topic, the reviewer strongly recommends to update the literature through read, discuss and cites in the references with great attention all of those recent interesting articles, that helps the authors to better introduce and discuss the aim of the study in light of the some factors related with others adhuvants, such as hyaluronic acid.

Response 1: Thank you for your suggestion. We added some lines and the relative citation about the hyaluronic acid at page 2 Line 63-66 to introduce other possible adjuvants in periodontal regeneration.

Point 2: The authors should be better specified, at the end of the introduction section, the rational of the study. In the material and methods section, should better clarify the outcome assessment and the period of screening of the articles.

Response 2: Thank you for your useful comment. We modified the end of the introduction to better explain the rationale of the present study. We modified the Material and Methods section specifying the period of screening of the articles.  Regarding the outcome assessment, the primary and secondary outcomes considered from the included study are extensively described at page 8, paragraph 2.7 “Summary measures”. However, to further clarify the outcomes assessed, we also added a sentence to paragraph 2.5 “Data extraction” to early present in the paper the outcomes researched.

Point 3: The conclusion should reinforce in light of the discussions.

Response 3: Thank you for your comment. We modified the Conclusion reinforcing the evidence of the results. The conclusion reads now as follows (page 16 Line 422-426): “Within its limits, PRF demonstrated better results than OFD alone in furcation treatment while its adjunction to BG seems to be less useful. However, more well-designed RCTs with low risk of bias are needed to clarify its potential role in combination with BGs.”

Point 4: Abstract: Better formulate the introduction section by better describe the background

Response 4: Thank you for your comment. Background section of the abstract was extended as requested giving more details about the rationale behind the use of PRF.

Point 5: Page 3, Line 71: please add the relative sentence

Response 5: We apologize but we could not understand what exactly the reviewer asked us to correct. That is why we have not carried out any change in the text.

Point 6: Please add a specific sentence that clarifies the results obtained in the first part of the discussion

Response 6: Thank you for your comment. We added the sentence at the end of the first part of the Discussion section as requested. This sentence reads in the text as follows (page 14, Line 341-343) “Hence, the application of PRF in the treatment of grade 2 furcation defects may be largely encouraged and preferred respect to the approach of a solely OFD.”

Point 7: Page 14 last paragraph of discussion: Please reorganize this paragraph that is not clear

Response 7: We re-organized the last part of the Discussion in order to better clarify the importance of not considering a mix of platelet concentrates due to their differences in handling and biological properties in writing a systematic review. This feature is the main factor that distinguishes our SR by some of the previous ones.